# Interaction of Varenicline with Classic Antiseizure Medications in the Mouse Maximal Electroshock-Induced Seizure Model

**DOI:** 10.3390/ijms24032616

**Published:** 2023-01-30

**Authors:** Piotr Bernat, Patrycjusz Kołodziejczyk, Jarogniew J. Łuszczki, Mirosław Zagaja, Piotr Tutka

**Affiliations:** 1Department of Experimental and Clinical Pharmacology, University of Rzeszów, 35-310 Rzeszów, Poland; 2Department of Occupational Medicine, Medical University of Lublin, 20-059 Lublin, Poland; 3Isobolographic Analysis Laboratory, Institute of Rural Health, 20-090 Lublin, Poland; 4Laboratory for Innovative Research in Pharmacology, University of Rzeszów, 35-310 Rzeszów, Poland; 5National Drug and Alcohol Research Centre, University of New South Wales, Sydney, NSW 2052, Australia

**Keywords:** antiseizure medications, carbamazepine, epilepsy, maximal electroshock, smoking cessation, varenicline

## Abstract

Varenicline (VAR) is a partial agonist of brain α4β2 nicotinic acetylcholine receptors recommended as a first line pharmacotherapy for smoking cessation. The aim of this study was to examine whether VAR affects the protective activity of four classic antiseizure medications, i.e., carbamazepine (CBZ), phenobarbital (PB), phenytoin (PHT), and valproate (VPA) on maximal electroshock (MES)-induced seizures, which may serve as an experimental model of human-generalized tonic-clonic seizures in mice. VAR administered intraperitoneally (i.p.) at a subthreshold dose of 0.5 mg/kg decreased the protective activity of CBZ against MES-induced convulsions, increasing its median effective dose (ED50) from 10.92 ± 1.0 to 18.15 ± 1.73 mg/kg (*p* < 0.01). The effect of VAR was dose-dependent because a lower dose of VAR (0.25 mg/kg) failed to antagonize the protective activity of CBZ. VAR administered at the subthreshold dose of 0.5 mg/kg had no impact on the protective activity of PB, PHT, and VPA in the mouse MES model. The inhibitory effect of VAR on the protective activity of CBZ against tonic-clonic convulsions most likely resulted from the pharmacodynamic mechanism(s) and was not associated with the changes in total brain concentrations of CBZ. VAR-evoked alterations in the anticonvulsive activity of CBZ may be of serious concern for epileptic tobacco smokers.

## 1. Introduction

Varenicline (VAR) and cytisine are the most effective drugs currently recommended for smoking cessation. VAR doubles the odds of quitting, compared to a placebo [1], and cytisine has been found at least as effective [2] or not to be noninferior to VAR [3]. The mechanism of their antismoking action is associated with a partial agonism of brain α4β2 nicotinic acetylcholine receptors (nAChRs) [4,5,6].

It has been demonstrated that the activation of brain nAChRs is involved in nicotine-induced clonic-tonic seizures in mice [7]. Both cytisine and VAR can cause convulsions in experimental animals. Cytisine induces clonic-tonic [8] and partial seizures in mice [9]. VAR administered in a single dose causes epileptic activity in rats [10].

VAR is generally a well-tolerated drug, but postmarketing surveillance suggests an association between VAR and increased risk of seizures [11,12,13]. Therefore, the U.S. Food and Drug Administration (FDA) updated the label for VAR in 2015 to warn about the increased risk of VAR-induced seizures. On the other hand, there is a study that did not find a significant association between increased risk of seizures and VAR use [14].

In our previous studies, cytisine administered intraperitoneally (i.p.) has been shown to reduce the anticonvulsive activity of some antiseizure medications (ASMs) in maximal electroshock (MES)-induced seizures [15] and the psychomotor seizure (6 Hz) test in mice [16]. The effect of cytisine on the action of these ASMs is unknown in epileptic patients. However, it has been concluded that cytisine-induced changes in the protection provided by some ASMs against convulsions can be of serious concern for epileptic smokers [15,16].

There have been no published studies investigating the effect of VAR on the anticonvulsant activity of ASMs. An understanding of these interactions and their mechanisms is important to the comprehensive management of epileptic smokers who have attempted to quit smoking using VAR. The aim of this study was to examine whether VAR affects the protective activity of four classic ASMs, i.e., carbamazepine (CBZ), phenobarbital (PB), phenytoin (PHT), and valproate (VPA) on MES-induced seizures, which may serve as an experimental model of human-generalized tonic-clonic seizures in mice. Additionally, the effects of VAR administered alone and its combinations with the ASMs on long-term memory, skeletal muscular strength, and motor impairment in mice were examined in the step-through passive avoidance task, the chimney test, and the grip-strength test, respectively.

## 2. Results

### 2.1. Effects of VAR on the Threshold for Tonic-Clonic Seizures in the MES-Induced Seizure Threshold Test

VAR administered i.p. at doses of 1 and 2 mg/kg significantly reduced the threshold for electrically induced seizures. VAR decreased the CS50 values from 8.1 mA (S.E.M. ± 0.43) for the control group to 6.52 ± 0.54 mA (*p* < 0.05) and 6.29 ± 0.48 mA (*p* < 0.05) for VAR 1 and 2 mg/kg, respectively. VAR at a dose of 0.5 mg/kg failed to change the seizure threshold (Table 1). The dose of VAR 0.5 mg/kg was determined to be the subthreshold dose.

### 2.2. Influence of VAR on the Anticonvulsant Activity of CBZ, PB, PHT, and VPA in the MES-Induced Seizure Test

VAR administered i.p. in a subthreshold dose of 0.5 mg/kg significantly increased the ED50 of CBZ from 10.92 ± 1.0 mg/kg to 18.15 ± 1.73 mg/kg (*p* < 0.01). When VAR was administered at a dose of 0.25 mg/kg along with CBZ, an increase of ED50 was not statistically significant (Table 2). VAR administered i.p. in the subthreshold dose of 0.5 mg/kg had no impact on the protective activity of PB, PHT, and VPA against MES-induced seizures (Table 2).

### 2.3. Long-Term Memory in Mice Treated with ASMs and VAR Alone or in Combination

A control did not show memory deficits challenged with the passive avoidance task. The same results were obtained in mice treated with VAR alone (0.5 mg/kg). VAR at the same dose administered with CBZ, PB, PHT, and VPA did not affect significantly retention times in experimental animals (Table 3).

### 2.4. Muscular Strength in Mice Pretreated with ASMs and VAR Alone or in Combination

A control group administered with vehicles had mean grip strength of 1.039 ± 0.046 N. When VAR was administered at the dose of 0.5 mg/kg, it was detected that the drug had no significant impact on skeletal muscular strength in the grip strength test. VAR applied at the dose of 0.5 mg/kg along with CBZ, PB, PHT, and VPA did not affect skeletal muscular strength in mice assessed in the grip strength test (Table 3).

### 2.5. Motor Coordination Performance among Mice Pretreated with ASMs and VAR Alone or in Combination

A control group of mice that received vehicles did not reveal an impairment on motor coordination in the chimney test (Table 3). VAR alone at the subthreshold dose of 0.5 mg/kg + vehicle for ASMs produced 12.5% of motor coordination impairment compared to the drug free group. VAR administered at a dose of 0.5 mg/kg with CBZ, PB, and PHT at their ED50 (18.15, 26.17 and 9.56 mg/kg, respectively) revealed 12.5% motor coordination impairment. When VAR at 0.5 mg/kg was combined along with VPA (370.8 mg/kg), motor impairment occurred in 37.5% of the tested mice (Table 3).

### 2.6. Influence of VAR on Total Brain Concentration of CBZ 

Total brain concentration of CBZ administered alone at the dose of 18.15 mg/kg (i.e., at the dose corresponding to its ED50 value from the MES test) was 2.84 ± 0.25 µg/g. VAR (0.5 mg/kg) coadministrated with CBZ (18.15 mg/kg) did not significantly change total brain concentration of CBZ (2.84 ± 0.25 µg/g for CBZ alone vs. 2.43 ± 0.31 µg/g for CBZ + VAR) (Figure 1).

## 3. Discussion

This is the first study that examined the effects of VAR, a commonly used smoking cessation aid, on the ability of four classic ASMs to protect against convulsions caused by MES, an experimental model of human-generalized tonic-clonic seizures in mice.

First, we demonstrated that single i.p. administration of VAR affected a threshold for seizures in the MES seizure threshold test. VAR at doses of 1 and 2 mg/kg significantly and dose-dependently reduced current strengths needed to provoke tonic-clonic seizures. This effect was not present at a dose of 0.5 mg/kg, which was applied in subsequent follow-up experiments as the subthreshold dose.

The principal finding in this study is that VAR at the subthreshold dose of 0.5 mg/kg decreased the protective activity of CBZ against MES-induced convulsions. VAR administered at this dose increased the ED50 of CBZ by 66%. The effect of VAR was dose-dependent because, as it was further observed, a lower dose of VAR (0.25 mg/kg) failed to antagonize the protective activity of CBZ. Interestingly, out of four ASMs tested in the study, a statistically significant interaction was observed between VAR and CBZ only. When VAR was applied in combination with PB, PHT, and VPA, there was no effect on the protective efficacy of PB, PHT, and VPA against MES-induced seizures.

VAR (Champix in Europe and Chantix in the U.S.) was approved by FDA in 2006 as a medicine to help quit tobacco smoking. Early postmarketing surveillance reports triggered a wide discussion as well as a call for an investigation targeted at the safety of VAR in neuropsychiatric patients. Signals emerging from the FDA Adverse Event Reporting System (FAERS) were linked to deterioration of depression, increased agitation, psychosis, or even suicidal thoughts or attempts [17]. As a consequence, the FDA issued a box warning on the product label to warn patients and clinicians about potential serious neuropsychiatric events [18]. However, the EAGLES study, which compared VAR with bupropion, nicotine replacement therapy, and a placebo and included more than 8000 participants, demonstrated that VAR was not associated with a significant increase in moderate-to-severe neuropsychiatric adverse events compared to the placebo [19]. In 2016, based on the EAGLES trial results the FDA decided to remove the warning box on VAR.

On the basis of the FAERS data, it was demonstrated in 2015 that patients treated with VAR had an increased risk of seizures [20]. The FDA reported 64 cases of seizures in patients using VAR. Half of the cases were among patients without previous history of seizures [21]. As a result of safety signals, the FDA implemented significant limitations in VAR use among epileptic patients. These concerns are important since patients with epilepsy are up to twice as likely to be tobacco smokers compared to those without epilepsy [22]. There are reports suggesting that a probability of a seizure attack in smokers with epilepsy is higher than among those who are not smokers [12]. These reports give a clear signal to the need for safe and effective antismoking therapy in smokers with epilepsy.

The nAChRs are involved in the addictive effects of nicotine [23]. The nAChRs are a main target for the antismoking action of VAR. The drug binds predominantly to the α4β2 subtype of nAChRs and acts by reducing the rewarding effect of nicotine, attenuating nicotine withdrawal, and decreasing cravings [5,24]. It was demonstrated that nAChRs are involved in the pathogenesis of some forms of epilepsy, i.e., nocturnal frontal lobe epilepsy [25,26]. Mutations of the α4 or β2 subunits alter functional properties of nAChRs by increasing sensitivity to acetylcholine [27]. It is known that nicotine induces clonic-tonic convulsions in animals through the excessive activation of nAChRs, leading to synaptic release of glutamate [28] and, as a consequence, to seizure activity [29]. Similarly, cytisine, another partial agonist of α4β2 nAChRs, was found to induce convulsions in animals [30,31] and humans [31]. Single systemic administration of cytisine caused a significant reduction of the seizure threshold for electroconvulsions in mice [15].

Theoretically, VAR and nicotine alone or both agents taken together, via their activation of nAChRs, might change the antiseizure activity of ASMs in smokers with epilepsy who continue to smoke while taking VAR [32]. In this study, VAR significantly reduced the protective activity of CBZ in the model of human-generalized tonic-clonic seizures. On the other hand, VAR did not affect the protective activity of three other ASMs: PB, PHT, and VPA. Thus, the interaction between VAR and classic ASMs is not general in nature, but it is limited only to one drug. 

Considering the presented results, it seems very likely that the inhibition of the protective activity of CBZ by VAR in the MES test may have a pharmacodynamic nature. There were no significant changes in total brain concentrations of CBZ following VAR administration. This suggests that the inhibitory effect of VAR is not related to pharmacokinetic factors. To explain the observed significant reduction of the anticonvulsant potency of CBZ after VAR administration, molecular mechanisms of action of both drugs should be borne in mind. In regards to CBZ, the drug binds to the alpha subunit of voltage-gated sodium channels, and, thus, it inhibits action potentials and decreases synaptic transmission in neurons [33,34]. At therapeutic concentrations, CBZ binds to adenosine A1 and A2 receptors [35] and inhibits NMDA-stimulated calcium influx in neurons [36]. Molecular mechanisms of action of VAR are linked to a full agonism on α7 nAChRs and a partial agonism on the α4β2, α3β4, and α6β2 subtypes of nAChRs [37,38]. VAR also displays a weak agonism on α3β2 nAChRs [37,39]. Considering the above-mentioned mechanisms, it seems that both drugs have no similar mechanisms that could mutually interfere and be responsible for the observed interaction. However, CBZ and VAR have a similar scaffold structure (Figure 2), and probably this is the reason that VAR can antagonize the anticonvulsive action of CBZ in the mouse MES model. It is highly likely that VAR can interact and allosterically displace CBZ from its binding sites, and, thus, the reduction in the antiseizure action of CBZ occurs. Although this hypothesis can readily explain the observed reduction in the anticonvulsant potency of CBZ after VAR coadministration, it needs experimental verification in further biomolecular studies.

It should be noted that VAR is not the only nAChRs agonist decreasing the anticonvulsive effects of ASMs. Nicotine antagonized the protective activity of topiramate in kainic acid-induced convulsions (a model for human temporal lobe epilepsy) through activation of nAChRs [40]. Cytisine injected i.p. was found to exert an inhibitory effect on the protective activity of PHT and lamotrigine against MES-induced seizures in mice [15]. Moreover, cytisine administered i.p. in a subthreshold dose of 2 mg/kg dose-dependently antagonized the beneficial actions of lacosamide, levetiracetam, and pregabalin in the six Hz-induced psychomotor seizures in mice [16]. The inhibitory effects of cytisine were not related to pharmacokinetic factors. 

VAR administered alone or in combination with CBZ did not affect long-term memory, skeletal muscular strength, and motor coordination as determined in the passive avoidance, the chimney, and the grip-strength tests, respectively. We can conclude that the interaction of VAR with CBZ relies exclusively on VAR’s impact on the anticonvulsant activity of CBZ without affecting its acute adverse effects. Of note, VAR given in combination with PB, PHT, and VPA also had no impact on long-term memory, skeletal muscular strength, and motor coordination of mice in the passive avoidance, the chimney, and the grip-strength tests, respectively.

Another fact requires explanation, since VAR was used in this study in doses of 0.25 and 0.5 mg/kg when combined with CBZ. In experimental pharmacology, it is difficult to directly extrapolate the doses of the studied drugs from animal experiments to clinical conditions; however, some algorithms allow approximate conversion of drug doses from animals to humans. Converting a dose of 0.5 mg/kg for VAR from mice to humans (according to the algorithm present by Nair and Jacob [41]), the human equivalent dose (HED) of VAR reaches 0.0405 mg/kg or 2.43 mg for a 60-kg adult human. Of note, the human doses for VAR range from 0.5 to 2 mg per day (approx. 0.02–0.04 mg/kg) [42]. Thus, VAR doses used in the study are comparable to those used in clinical conditions, which provides us with a general reference to the activity of the aforementioned drug applied in vivo.

The main limitation of this study is that an influence of VAR on the protective activity of the ASMs was investigated in only one seizure animal model. In order to draw the right and full conclusion, many acute and chronic models of epilepsy should be used to avoid false predictions.

## 4. Materials and Methods

### 4.1. Animals and Experimental Conditions

The study was conducted on adult Swiss mice weighing in a range of 24 ± 3 g. The animals were kept in standard conditions with unlimited access to chow pellets, water, and appropriate bedding in the cages. During the experiments, stable ambient conditions were kept, e.g., 18–22 °C along with 52–58% relative humidity. Natural day and night rhythm was ascertained with a 1-week-long acclimatization period. Experimental groups were taken randomly and ranged from 8 to 24 mice. The tests were performed between 8:00 and 15:00. The animals were used only once. Control groups were always tested on the same day as corresponding experimental groups. Total number of animals used in this study was 408.

### 4.2. Drugs

The following ASMs were used in the experiments: carbamazepine (CBZ), phenobarbital (PB), phenytoin (PHT), and valproate (VPA). (All the drugs were purchased from Sigma-Aldrich, St. Louis, MO, USA.) All ASMs were administered (i.p.) as follows: PHT—120 min, PB—60 min, CBZ and VPA—30 min prior to the MES test. Varenicline (VAR, Pfizer Ltd., Sandwich, UK) was administered i.p. 30 min before the MES test. The route of i.p. administration of VAR and the pretreatment time before testing of its anticonvulsant and side effects were based upon information from our pilot study. The same drugs’ application time limits were applied to the passive avoidance test, the grip strength test, and the chimney test. All drugs were suspended in a 1% solution of Tween 80 (Sigma-Aldrich, St. Louis, MO, USA) in distilled water, except for VPA, which as an acid was dissolved in distilled water only. Fresh solutions were prepared every day at a volume 5 mL/kg of body weight. Control animals were given equivalent amounts of sterile distilled water or 1% solution of Tween 80 via corresponding route.

### 4.3. Electrically-Induced Seizures

Tonic-clonic seizures were provoked by current delivered through ear-clip electrodes by the rodent shocker generator (constant-current stimulator Type 221, Hugo Sachs Electronik, Freiburg, Germany; stimulus duration 0.2 s) according to [43]. All animals were acclimatized to the experimental conditions at least 30 min before the test. The criterion for convulsant activity was tonic hind limb extension (i.e., the hind limbs of animals became extended at 180° to the plane of the body main axis).

### 4.4. MES Seizure Threshold Test

MES seizure threshold test was performed for four VAR doses: 0.25, 0.5, 1, and 2 mg/kg. Current simulation lasted for 0.2 s with different intensities (4–10 mA). Each time the number of convulsing out of total mice in an experimental group was registered and a dose-response curve was established through [44] test with subsequent calculations of median current strength 50 (CS50; in mA). Each CS50 indicates current intensity necessary to provoke tonic seizures represented by hind limbs extension in 50% of animals.

### 4.5. MES Seizure Test

Mice were subjected to the current with constant intensity (25 mA) and stimulus duration (0.2 s). The experimental endpoint was complete protection against tonic convulsions during 1 min of observation. Control groups received progressive doses of ASM and vehicle. All animals in control groups produced seizures. To determine median effective doses (ED50 values) of ASMs, the drugs were administered at the following doses: CBZ 8, 10, 14, 16, 18, 20, and 25 mg/kg; PB 20, 25, 30, and 35 mg/kg; PHT 8, 10, and 14 mg/kg; and VPA 300, 350, 400, and 450 mg/kg. In order to determine the effect of VAR on the anticonvulsive activity of the ASMs, the study groups received progressive doses of the ASMs in combination with VAR at the dose of 0.5 mg/kg, i.e., the highest studied dose that did not significantly change the threshold in the MES seizure threshold test. If a significant effect of VAR on effective dose (ED50; dose of the drug protecting 50% of animals against MES) of the ASM was found, then a lower dose of VAR (0.25 mg/kg) was administered following the same experimental procedure as for a subthreshold dose of VAR. The ED50 of the ASM for mice pretreated with VAR was compared with the respective ED50 of the ASM administered separately (+vehicle).

### 4.6. Passive Avoidance Task

Influence of VAR alone (0.5 mg/kg) and its combination with CBZ, PB, PHT, and VPA (at doses corresponding to their ED50 values from the MES test) on long-term memory was quantified by the step-through passive avoidance task using methodology described elsewhere [45]. A procedure was performed with placement of the animals in illuminated box (10 cm × 13 cm × 15 cm) that had access to a dark compartment (25 cm × 20 cm × 15 cm) easily accessible to the animals. The dark compartment was equipped with an electric grid floor. Entrance of mice to the dark compartment (natural behavior of rodents) was punished by an adequate electric foot shock impulse (0.6 mA for 2 s). The animals that did not enter the dark compartment were excluded from subsequent experiments. Twenty-four hours later, the pretrained mice were placed again into the illuminated box and observed for up to 3 min. Results were calculated as median latencies (retention times in s) with 25th and 75th percentiles. 

### 4.7. Grip Strength Test

The effects of VAR alone (0.5 mg/kg) and its combination with CBZ, PB, PHT, and VPA were tested in the skeletal muscular grip strength test [46]. Applied doses of the ASMs derived from the MES test (ED50). Measurement of front limb muscle strength was facilitated by BioSeb apparatus (Chaville, France). The apparatus is an electronic dynamometer connected with a wire grid (8 cm × 8 cm) capable of measuring muscle strength of the animals in newtons (N). An animal was uplifted by the tail with ability to catch the grid by front limbs. Once a grip was established, the animal was dragged backwards by the tail until the grip was released [47]. The maximal force exerted by the animal before losing the grip was recorded. The skeletal muscular strength in mice was expressed in N as means ± standard error (S.E.M.) of 8 determinations. 

### 4.8. Chimney Test

A chimney test was applied in order to check motor coordination [48]. Mice were treated with VAR alone (0.5 mg/kg) or VAR in combination with the ASMs at ED50s doses derived from the MES test. The experiment used a transparent plastic tube (30 cm length and 3 cm inner diameter). The inner surface of the tube allowed proper grip of the animal limbs when the tube was put to the vertical position. The natural behavior of mice is to go out backwards from the tube when it changing its position from horizontal to vertical. An animal was placed into the tube, and once the animal showed up at the end of the tube, an operator changed its position and measured time (in seconds) until the animal showed up on top of the tube. Results are presented as percentage of the animals with impairment of motor coordination (the inability of the animals to perform the test within 60 s).

### 4.9. Measurement of Total Brain Concentrations of CBZ

Pharmacokinetic estimation of total brain ASM concentrations was performed only for combinations of VAR with CBZ for which the anticonvulsant effect in the MES seizure test was significantly greater than for control animals. In order to understand if pharmacokinetic interaction between CBZ and VAR was present, measurement of total brain concentrations of CBZ was conducted. The measurement was performed in the following way: each animal pretreated with CBZ alone or in combination with VAR was decapitated at time reflecting the peak of maximum anticonvulsant effect of CBZ in the MES test; subsequently a whole brain was removed from the skull, weighed, harvested, and finally homogenized with usage of Abbott buffer (1:2 weight/volume). The homogenates were centrifuged at 10,000× *g* for 10 min with usage of Centrifuge MPW 350 (MPW Med. Instruments, Warsaw, Poland). Supernatant obtained that way was analyzed with the fluorescence polarization immunoassay (FPIA) method and with Abbott Architect c4000 analyzer (Abbott Park, IL, USA). A manufacturer’s supplied kit for detection of CBZ was applied. Total brain concentration of CBZ (in µg/g of wet brain tissue) was measured as means ± S.E.M. of 8 samples with CBZ brain preparations as reference and 8 brain samples with CBZ+VAR preparations to exclude pharmacokinetic relationship. 

### 4.10. Statistical Methods

The CS50 and ED50 values were calculated by log-probit analysis [44] and statistically analyzed using one-way analysis of variance (ANOVA) followed by the post-hoc Tukey–Kramer test for multiple comparisons. Total brain concentrations of CBZ given alone and in combination with VAR were statistically compared using the unpaired Student’s t test. The data from the passive avoidance task were verified with Kruskal–Wallis nonparametric ANOVA. The results from the grip strength test were analyzed with one-way ANOVA followed by the post-hoc Dunnett’s multiple comparison test. The Fisher’s exact probability test was used to analyze the results from the chimney test. The index of probability less than 0.5 (*p* < 0.05) was considered significant in the comparative analysis.

## 5. Conclusions

The effect of VAR on the anticonvulsive activity of the classic ASMs in MES-induced seizures in mice was limited to only CBZ but not to PB, PHT, and VPA. It seems that the inhibitory effect of VAR on the protective activity of CBZ against tonic-clonic convulsions most likely resulted from pharmacodynamic mechanism(s) and was not associated with the changes in total brain concentrations of CBZ. The mechanism of the pharmacodynamic interaction between VAR and CBZ should therefore be further investigated. Combination of VAR with CBZ, PB, PHT, and VPA had no effect on long term memory, skeletal muscular strength, and motor coordination in mice. 

VAR-evoked alterations in the anticonvulsive activity of CBZ, the mainstay of the pharmacological management of focal and generalized tonic-clonic seizures for many years and a current first-line therapy for partial-onset seizures [49], can be of serious concern for epileptic tobacco smokers. Reduction of the anticonvulsant effect of CBZ may increase the risk of seizure attacks. Our findings recommend further research on VAR in respect its influence on the anticonvulsive activity of CBZ.

## Figures and Tables

**Figure 1 ijms-24-02616-f001:**
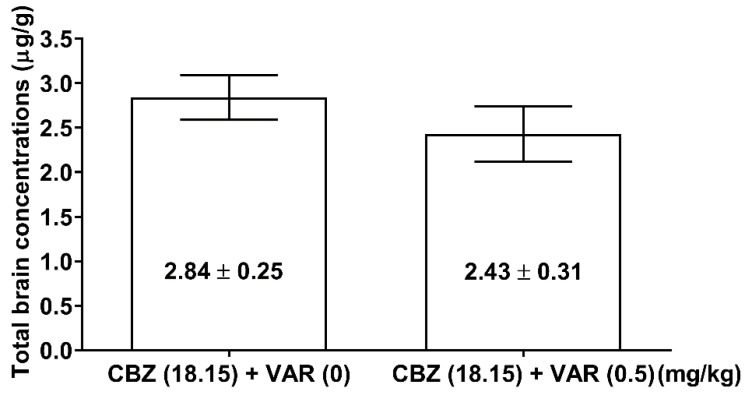
Effect of varenicline (VAR) on total brain concentrations of carbamazepine (CBZ) in mice. Total brain concentrations of CBZ as means (±S.E.M.) of 8 determinants were assayed with fluorescence polarization immunoassay (FPIA) and expressed in µg/g of wet brain tissue.

**Figure 2 ijms-24-02616-f002:**
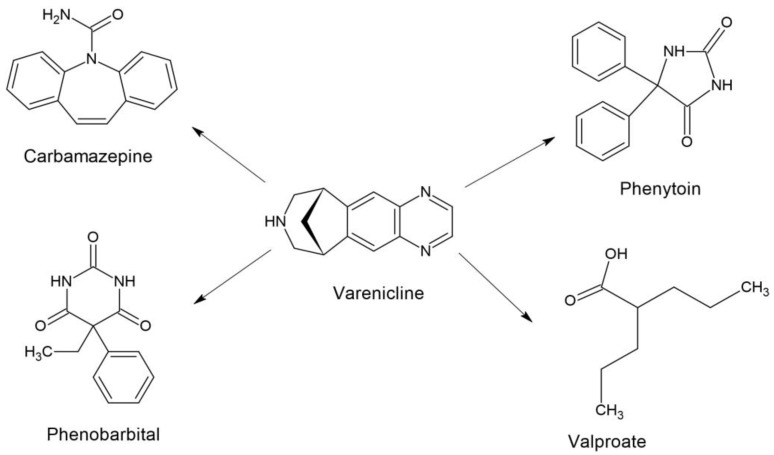
Chemical structures of the studied drugs.

**Table 1 ijms-24-02616-t001:** Effect of varenicline (VAR) on the threshold for tonic-clonic seizures in mice.

Treatment (mg/kg)	CS50 (mA)	*n*
VAR (0)	8.10 ± 0.43	24
VAR (0.25)	8.32 ± 0.48	16
VAR (0.5)	7.45 ± 0.45	16
VAR (1.0)	6.52 ± 0.54 *	24
VAR (2.0)	6.29 ± 0.48 *	16

Results are presented as median current strengths (CS50 in mA ± S.E.M.) required to produce tonic hind limb extension in 50% of animals tested. *n*—number of animals tested at those current strength intensities for which seizure effects ranged between 16% and 84%. Statistical analysis of data for multiple comparisons was performed with one-way ANOVA followed by the post-hoc Tukey–Kramer’s test (F (4;91) = 3.405; *p* = 0.0121); * *p* < 0.05 vs. control group (VAR (0)-treated animals).

**Table 2 ijms-24-02616-t002:** Effect of varenicline (VAR) on the protective activity of classic ASMs against MES-induced seizures in mice.

Treatment (mg/kg)	ED50 (mg/kg)	*n*
CBZ + VAR (0)	10.92 ± 1.00	24
CBZ + VAR (0.25)	15.73 ± 0.85	8
CBZ + VAR (0.5)	18.15 ± 1.73 **	24
PB + VAR (0)	27.71 ± 1.78	16
PB + VAR (0.5)	26.17 ± 2.07	16
PHT + VAR (0)	11.27 ± 1.23	16
PHT + VAR (0.5)	9.56 ± 1.06	16
VPA + VAR (0)	337.2 ± 28.9	24
VPA + VAR (0.5)	370.8 ± 21.0	24

Results are presented as median effective doses (ED50 in mg/kg ± S.E.M.) of ASMs, protecting 50% of animals tested against MES-induced seizures in mice. *n*—total number of animals used at those doses whose anticonvulsant effects ranged between 4th and 6th probit. Statistical analysis of data for multiple comparisons was performed with one-way ANOVA followed by the post-hoc Tukey–Kramer’s test (F (2;53) = 7.536; *p* = 0.0013); ** *p* < 0.01 vs. control (CBZ+VAR (0)-treated) animals.

**Table 3 ijms-24-02616-t003:** Influence of varenicline (VAR) alone and its combination with four classic ASMs on skeletal muscular strength, long-term memory, and motor coordination in mice.

Treatment (mg/kg)	Muscular Strength (N)	Retention Time (s)	Motor Coordination Deficits (%)
Vehicle + vehicle	1.039 ± 0.046	180 (180; 180)	0
VAR (0.5) + vehicle	1.082 ± 0.028	180 (180; 180)	12.5
VAR (0.5) + CBZ (18.15)	1.068 ± 0.030	180 (180; 180)	12.5
VAR (0.5) + PB (26.17)	1.046 ± 0.044	180 (180; 180)	12.5
VAR (0.5) + PHT (9.56)	1.054 ± 0.036	180 (180; 180)	12.5
VAR (0.5) + VPA (370.8)	1.029 ± 0.034	175.5 (155.7; 180)	37.5

Table data represents doses of the studied drugs corresponding to the ED50 values of classic ASMs from the tonic-clonic seizure model. Results are presented as: mean muscular strengths (in newtons [N] ± S.E.M.) in mice from the grip-strength test; median retention times (with 25th and 75th percentiles in parentheses) of the mice from the passive avoidance task; and percentage of animals with impairment of motor coordination from the chimney test, respectively.

## Data Availability

Data are contained within the article.

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
