# Peer review of "Interaction of Varenicline with Classic Antiseizure Medications in the Mouse Maximal Electroshock-Induced Seizure Model"

_ijms, 2023, doi:10.3390/ijms24032616_

Round 1

Reviewer 1 Report

The manusript "Interaction of varenicline with classic antiseizure medications 2 in the mouse maximal electroshock-induced seizure model" is interesting and well written. I have some suggestions:

1. Abstract - line 27 "... the anticonvulsive activity of CBZ can be of serious" the term may should be used instrad of can because the research was performed in animal models and not in humans.

2. Table  2. I suggest you omit the F and p value in the table - you can put it in legend because it is difficult to follow the rows in the present table.

 3. Discussion - could you please discuss more in detail the interactions between VAR and CBZ? You say in line 200 that it is difficult to explain the mechanism of the interaction because it was not observed with other medications. However, there must be something specific in the mechanism of action of CBZ which may explain your observation. I agree that the mechanism is probably pharmacodynamic since varenicline is not a substrate, inducer or inhibitor of CYP450 isoensymes and the concentration of CBZ was unchanged. 

Author Response

REV_1

The manusript "Interaction of varenicline with classic antiseizure medications 2 in the mouse maximal electroshock-induced seizure model" is interesting and well written. I have some suggestions:

1. Abstract - line 27 "... the anticonvulsive activity of CBZ can be of serious" the term may should be used instrad of can because the research was performed in animal models and not in humans.

Reply:

The word “can” has been replaced with “may” as suggested.

2. Table  2. I suggest you omit the F and p value in the table - you can put it in legend because it is difficult to follow the rows in the present table.

Reply:

The F statistics and p values have been removed from the tables 1 and 2 and placed in the legend to tables 1 and 2, as requested.

3. Discussion - could you please discuss more in detail the interactions between VAR and CBZ? You say in line 200 that it is difficult to explain the mechanism of the interaction because it was not observed with other medications. However, there must be something specific in the mechanism of action of CBZ which may explain your observation. I agree that the mechanism is probably pharmacodynamic since varenicline is not a substrate, inducer or inhibitor of CYP450 isoensymes and the concentration of CBZ was unchanged. 

Reply:

The interaction between VAR and CBZ has been discussed in more details as requested. We have added a novel Figure 2 to explain possible mechanisms.

Reviewer 2 Report

This manuscript discussed the effects of varenicline on the anticonvulsive activity of four anticonvulsants in maximal electroshock-induced seizures in mice. Overall, the study is well designed, methods are appropriate, the manuscript is well written, and the novel and significant results in the field of pharmacology of CNS drugs have been obtained.

The use of only one animal model for evaluating the influence of varenicline on the protective activity of antiseizure medications was stated as the main limitation of this study. However, in my opinion, there are other concerns that should be clarified before the publication.

First, the choice of varenicline doses in the experiments is disputable. Varenicline was applied in doses of 0.25 - 2 mg/kg in mice. How were these doses chosen? The human doses for this drug range from 0.5 to 2 mg per day (approx. 0.02-0.04 mg/kg). Did applied doses in this study had clinically relevant concentrations in mice as a result? This needs to be discussed in the manuscript.

Second, why did some experimental groups have 8 mice, and the others 24? Was ANOVA test the most appropriate for your results considering the varying number of samples in different experimental groups?

Besides, it was stated in the manuscript that (all) drugs were suspended in a 1% solution of Tween 80. Why did you add Tween 80 if the drugs remained suspended and not dissolved?

Author Response

REV_2

This manuscript discussed the effects of varenicline on the anticonvulsive activity of four anticonvulsants in maximal electroshock-induced seizures in mice. Overall, the study is well designed, methods are appropriate, the manuscript is well written, and the novel and significant results in the field of pharmacology of CNS drugs have been obtained.

The use of only one animal model for evaluating the influence of varenicline on the protective activity of antiseizure medications was stated as the main limitation of this study. However, in my opinion, there are other concerns that should be clarified before the publication.

  1. First, the choice of varenicline doses in the experiments is disputable. Varenicline was applied in doses of 0.25 - 2 mg/kg in mice. How were these doses chosen? The human doses for this drug range from 0.5 to 2 mg per day (approx. 0.02-0.04 mg/kg). Did applied doses in this study had clinically relevant concentrations in mice as a result? This needs to be discussed in the manuscript.

Reply:

The doses of VAR and a problem of translating doses of the drug from animals to humans has been discussed, as suggested.

  1. Second, why did some experimental groups have 8 mice, and the others 24? Was ANOVA test the most appropriate for your results considering the varying number of samples in different experimental groups?

Reply:

Number of animals in Tables are presented only for those groups of animals for which the observed effects ranged between 4th and 6th probits or 16% and 84% (according to Litchfield and Wilcoxon, 1949). If the observed anticonvulsant effects are lower than 16% (i.e., 12.5% - 1 out of 8 animals was protected from MES-induced seizures) or higher than 84% (i.e., 87.5% - 7 out of 8 animals was protected from seizures) we cannot present these groups of animals in tables. This is why the numbers of animals differ in Tables 1 and 2.

One-way ANOVA is the appropriate test to statistically analyze three or more data (in the MES threshold test and when analyzing CBZ data). The Tukey-Kramer post-hoc test is used if variances of the analyzed means differ. The varying number of animals in different experimental groups may result in varying variances. This is why, the Tukey-Kramer’s post-hoc test was used in this study.

  1. Besides, it was stated in the manuscript that (all) drugs were suspended in a 1% solution of Tween 80. Why did you add Tween 80 if the drugs remained suspended and not dissolved?

Reply:

Since varenicline, PB, PHT and CBZ are insoluble in water, the drugs must be suspended in a 1% aqueous solution of Tween 80 so as to prepare the drugs for i.p. injection. In experimental pharmacology, Tween 80 is widely used as a suspending agent for drugs insoluble in water. 

Round 2

Reviewer 2 Report

The manuscript is improved after revisions.